# GPS displacement dataset for study of elastic surface mass variations

Athina Peidou[1], Donald Argus[1], Felix Landerer[1], David Wiese[1] and Matthias Ellmer[1]

Jet Propulsion Laboratory, California Institute of Technology, Pasadena, CA, USA, 2023

*Correspondence to*: Athina Peidou (athina.peidou@jpl.nasa.gov)

## Abstract

Quantification of uncertainty in surface mass change signals derived from GPS measurements poses challenges, especially when dealing with large datasets with continental or global coverage. We present a new GPS station displacement dataset that reflect surface mass load signals and their uncertainties. We assess the structure and quantify the uncertainty of vertical land displacement derived from 3045 GPS stations distributed across the continental US. Monthly means of daily positions are available for 15 years. We list the required corrections to isolate surface mass signals in GPS estimates and screen the data using GRACE(-FO) as external validation. Evaluation of GPS timeseries is a critical step, which identifies a) corrections that were missed; b) sites that contain non-elastic signals (e.g., close to aquifers); and c) sites affected by background modelling errors (e.g., errors in the glacial isostatic model). Finally, we quantify uncertainty of GPS vertical displacement estimates through stochastic modeling and quantification of spatially correlated errors. Our aim is to assign weights to GPS estimates of vertical displacements, which will be used in a joint solution with GRACE(-FO). We prescribe white, colored and spatially correlated noise. To quantify spatially correlated noise, we build on the common mode imaging approach adding a geophysical constraint (i.e., surface hydrology) to derive an error estimate for the surface mass signal. We study the uncertainty of the GPS displacement timeseries and find an average noise level between 2-3 mm when white noise, flicker noise, and RMS of residuals about a seasonality and trend fit are used to describe uncertainty. Prescribing random walk noise increases the error level such that half of the stations have noise > 4 mm, which is systematic with the noise level derived through modeling of spatial correlated noise. The new dataset is suitable for use in a future joint solution with GRACE(-FO)-like observations.

Keywords:  GPS uncertainty | elastic displacement | GRACE-FO | surface mass change

## 1. Introduction

For more than two decades, the Gravity Recovery and Climate Experiment (GRACE) space gravity mission and its nearly identical successor mission, GRACE-Follow on (GRACE-FO), have provided mass change estimates through tracking the time-variable part of the Earth's gravity field (Landerer et al., 2020). Mass change products are typically given on a monthly basis and have been used to study a variety

of critical climate-related factors (Tapley et al., 2019), such as sea level rise (Frederikse et al., 2020); ice
mass change (Velicogna et al., 2020); prolonged drought periods (Thomas et al., 2014) and regional flood
potentials (Reager et al., 2014). The measurement geometry of GRACE(-FO) limits the study of
geophysical processes to spatial scales of ~300 km and larger, for monthly timespans. Recent community
reports (Pail et al., 2015, Wiese et al., 2022) have highlighted the utility and need of mass change
observations at improved spatial resolutions to address a number of science and applications objectives.
Examples include closure of the terrestrial water budget for small to medium sized river basins, and
separation of surface mass balance from ice dynamic processes at the scale of individual outlet glacier
systems.
The spatial resolution of gravity maps derived from satellite measurements is limited by sampling at
altitude. Fusion with external geodetic data sources, however, can improve spatial resolution over what
can be achieved only with satellite gravimetry. GPS position timeseries have been used widely to study
the elastic response of Earth's surface to mass loading (e.g., Argus et al., 2017; Fu and Freymueller,
2012) and can provide information at short wavelengths (~100km) (Argus et al., 2021).  Solid Earth
responds elastically to changes in the surface load of water, snow, ice, and atmosphere. When the Earth's
surface is loaded with mass (e.g., snow and water) it subsides; and when mass loads are removed the
surface rises. Thus, the Earth's response follows the water cycles such that: precipitation and snow
accumulation cause subsidence of the surface and snow melt, evaporation and water run off allow the
Earth's surface to bounce back (uplift). Focus is typically placed on the radial direction (vertical), due to
the rapid decrease of vertical displacement with the distance from a surface load (Argus et al., 2017),
which leads to high fidelity estimates in the space domain. Note that across certain geological formations
such as aquifers, subduction zones and regions with volcanic activity surface loading is mixed with other
solid Earth/geophysical processes making it difficult to isolate the elastic component. Therefore, GPS
sites located at the vicinity of such formations are omitted.
GPS displacements between two epochs have many different signals embedded in them; i.e., those related
to non-tidal atmospheric and oceanic loading, solid Earth phenomena such as tectonics, glacial isostatic
adjustment, and others related to surface mass changes.  With the proper treatment (see Section 2) GPS
stations can capture local surface mass changes. We are interested in isolating the signals that reflect the
Earth's elastic response to mass variations, thus we apply a set of corrections to GPS vertical
displacement estimates, and then we screen the data for outliers or potential errors. The data screening
process checks for consistency between GPS and GRACE(-FO) vertical displacement estimates (similar
analysis has been performed by Yin et al., 2020; Blewitt et al., 2001; van Dam et al., 2001; Becker and
Bevis, 2004; Davis, 2004; Tregoning et al., 2009; Tsai, 2011 and Chew et al., 2014) and identifies outliers
that statistical tests fail to pick up (He et al., 2018).
The last step is to estimate uncertainty in the screened dataset. Since our purpose is to isolate surface mass
load signals, we define *error* as any vertical displacement signal that does not reflect an elastic surface
mass load. The reported uncertainty reflects the sum of all error sources to the measurement and is the
final product of this study. Error correlation (temporal and spatial) and the deficiency of stochastic noise
models to describe the error realistically are the main challenges in this uncertainty quantification task.
Error sources include errors driven by satellite antenna phase centre offsets (Haines et al., 2004;
Santamaria-Gomez et al., 2012); atmospheric pressure models (Kumar et al., 2020); non-tidal ocean
loading (Jiang et al., 2013); satellite orbits (Ray et al., 2008; Amiri-Simkooei ,2013); earth orientation
parameters (Rodriguez-Solano et al., 2014); and tectonic trends and post-seismic relaxation after
earthquake activity (Ji and Herring, 2013; Crowell et al., 2016).
The GPS position timeseries have common mode displacements [Tian and Shen 2016], including both a
common mode error strongly varying each day and a common mode signal associated with seasonal water
fluctuations. Wdowinski et al. (1997) first defined common mode error to be a series of rigid-body
translations that reflect an error in the position of all geodetic sites in an area relative to an absolute
reference frame; by removing the mean position (or stack) of all sites in an area, scientists recover more
accurate estimates of relative position contained in the data.  Dong et al. (2006) and Serpelloni et al.
(2013) defined common mode error in a more sophisticated manner using principal or independent
component analysis such that they remove spatially correlated, temporally incoherent error. Independent
is different than principal component analysis in that it finds the maximum independence of the
components instead of minimum correlation (Milliner et al., 2019; Liu et al., 2015). Common mode
displacements includes both error (such as that associated with error in satellite orbits) and signal (such as
the seasonal oscillation of elastic vertical displacement in elastic response to seasonal fluctuations in mass
between the hemispheres) (Sun et al. 2016).
Considering the increased number of GPS stations and the limitations posed by the existing
methodologies, Kreemer and Blewitt (2021) used a robust methodology to estimate the common spatial
components of GPS residuals (i.e., the remaining signals of a timeseries after subtraction of a trajectory
model). A trajectory model is a model consisting of an offset, a rate, and a sinusoid with a period of 1
year (Bevis and Brown, 2014). The so-called common mode component (CMC) imaging technique was
originally introduced by Tian and Shen (2016) and quantifies the spatial correlation of the residuals
(position or vertical displacement timeseries anomaly with respect to a trajectory model) of unequal-
length timeseries using information from neighbor stations. It is important to note that CMC reflects both
spatially correlated noise and spatially correlated signals, including elastic displacements, that a trajectory
model fails to describe.
Spectral analysis of the residuals (with respect to a trajectory model, see Eq.2) is an alternative way to
estimate the noise level of vertical displacement series for each GPS station. The spectrum of the
residuals can be approximated by white or colored noise (flicker, random walk, power law approximation,
generalized gauss markov etc.), or by a combination of white and colored noise (Williams et al., 2004;
Bos et al., 2008; Klos et al., 2014). A summary of the different noise models and their power distribution
can be found in He et al. (2018). Several standard GPS timeseries analysis packages are available to
perform such an analysis, e.g., the Create and Analyze Timeseries (CATS) (Williams, 2008) and Hector
(Bos et al. 2013). Various studies in the past suggested that the residuals are better described by a
combination of white and flicker noise (see e.g., Klos et al., 2014; Argus et al., 2017), with the latter
contributing the most (Argus and Peltier, 2010). Recently, Argus et al. (2022), showed that the longer the
timeseries the more the spectrum of GPS residuals converges with the noise model of random walk.
Here, we outline a comprehensive framework for processing large datasets (continental and/or global) of
GPS timeseries, to derive estimates that only reflect surface mass signals, for use in a joint inversion with
GRACE(-FO) measurements. Originally, we layout the corrections required to capture local surface mass
changes (Section 2.1). Our interest is to make the process as automated as possible, thus we set a number
of evaluation metrics to detect outliers among all candidate (for the joint inversion) sites. Stations flagged
as outliers are further evaluated for extra corrections (e.g., offsets; poor site maintenance etc.). Finally, we
assign weights to each GPS vertical displacement record. We test the most popular methodologies to
quantify the error, considering time-correlation, spatial-correlation and/or white noise (Section 3). Note
that for spatially correlated noise the commonly used PCA/ICA is not as applicable to our use case,
because our dataset extends over very large spatial areas (continental). CMC imaging (Kreemer and
Blewitt; 2021) fits our needs better. We build on the existing CMC algorithm to remove hydrology
signals from the error estimate by deriving surface loading signals from a hydrology model and removing
them from the GPS vertical displacements (see Section 3 for more details). The final product is a new
dataset with GPS vertical displacement estimates that reflect elastic mass variations and their
uncertainties.
**2. GPS data processing and screening**
2.1 Isolating surface mass loading fingerprint from GPS vertical displacements
We analyze positions of 3054 GPS sites as a function of time from 2006 to 2021 estimated by scientists at
the Nevada Geodetic Laboratory (NGL) (Blewitt et al. 2018).  Technologists at Jet Propulsion Laboratory
(JPL) first estimate satellite orbits, satellite clocks, and positions for a core set of roughly 50 sites on
Earth's surface (Bertiger et al. 2020). NGL uses JPL's clock and orbit products and performs point
positioning to a total of about 18,500 GPS sites distributed across the world.  Following the International
Earth Rotation Standards (IERS) (Petit and Luzum, 2012) NGL's positions are corrected for solid Earth,
ocean, and pole tides.  NGL's positions in International Terrestrial Reference Frame 2014 (ITRF2014)
(Altamimi et al. 2016) are more accurate than NGL's previous estimates of positions in ITRF2008.  NGL
estimates GPS wet tropospheric delays each day using the ECMWF weather model (Simmons et al. 2007)
and the VMF1 tropospheric mapping function (Boehm et al. 2006).  We input the NGL position
timeseries, derive the displacement relative to a reference epoch and then follow Argus et al. (2010, 2017,
2021) to isolate the part of GPS displacements reflecting solid Earth's elastic response:
a. Construct timeseries of elastic displacement uninterrupted by offsets due to antenna substitutions or
earthquakes that pass through a specific reference time (such as Jan 1, 2014) by eliminating data before
and /or after an offset.
b. Identify and omit GPS sites recording primarily i. poreoleastic response to change in groundwater, ii.
strong volcanic fluctuations, and iii. postseimic transients following Argus et al. (2014, 2017, 2022).  In
the west U.S., GPS sites responding to groundwater change have maximum height around April when
water is maximum, subside in the long term faster than 1.8 mm/yr, exhibit strong transients, and/or are
located in known aquifers (Argus et al. 2014).  Volcanic activity is readily identified by Interferometric
Synthetic Aperture Radar (InSAR) and GPS observations of strong transients and anomalous sustained
uplift or subsidence (Argus et al. 2014, Hammond et al. 2016).
c. Remove non-tidal atmospheric (NTAL) and non-tidal oceanic (NTOL) mass loading by interpolating
global grids of elastic displacements calculated by the German Center for Geoscience (GFZ) (Dill
Dobslaw, 2013) following the method of Martens et al. (2020).
d. Remove glacial isostatic adjustment as predicted by model ICE-6G_D (VM5a) (Peltier et al. 2015,
2018; Argus et al. 2014).
e.  Remove interseismic strain accumulation associated with locking of the Cascadia subduction zone
using an upgrade of the model of Wang et al. (2018).  The model is superposition of 2/3 of the elastic and
1/3 of the viscoelastic model of Wang et al. (2018).  We communicated with Li Wang and his team at
National Resources Canada, that the Wang et al. (2018) model does not fit the available GPS data; they
have produced an interim model using our input that more nearly fits the GPS data.
f.  Average the daily estimates of GPS vertical displacements into monthly means centered at the center of
each month from January 2006 to June 2021.
To compare GPS with GRACE(-FO) vertical displacement estimates we reference the series to the epoch
with the most GPS site records, which is September 2012. This process results in an 11% loss of stations
(i.e., no available measurement on 09/2012). Similar to Yin et al. (2020), detrended monthly estimates of
each station that are larger than 3σ relative to the mean of the timeseries are considered outliers and
removed from the dataset. Statistical outliers comprise ~0.5% of the records.
2705 (or 88.8%) of GPS stations remain after the choice of reference epoch, the 3σ test and the removal
of sites with non-elastic loading response. The distribution of sites is denser along the East and West
coasts, and fairly sparse in the central-north US (Fig.1). Series of two arbitrary stations (HIVI and NJWT)
located at the West and East coast respectively, are shown in Fig. 1. The response of the Earth on the
extensive drought period in California between 2011.5-2015.5 is captured in the uplift trend mapped by
HIVI station (Fig.1, top right panel; dashed blue line).

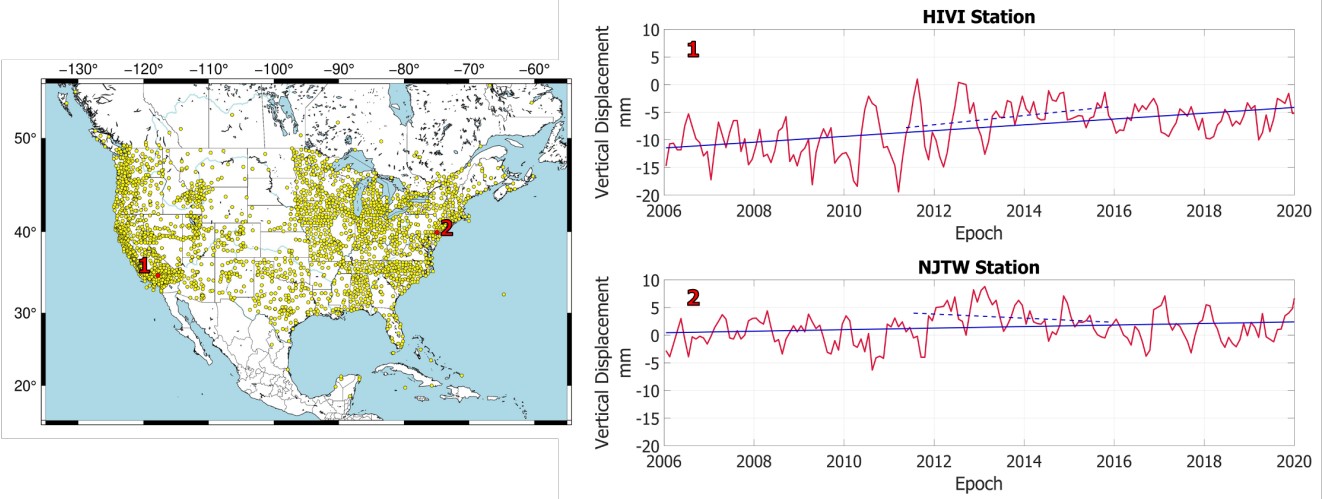

Figure 1: Left panel) Map of study area. GPS stations are shown in yellow; Right panel) Vertical
displacement timeseries of two random stations (red line). Solid blue line denotes the overall trend of the
timeseries and dashed blue line the trend between (2011.5-2015.5). Note the significant uplift of the HIVI
station located in southern California.
2.2 External validation datasets - Time-variable gravity field
We compare GPS observations of vertical displacement against GRACE(-FO) estimates of solid Earth's
elastic vertical displacement from terrestrial water, snow, and ice.
To compare to GRACE(-FO), we analyze JPL's three-degree mascon solution (Release 6, Watkins et al.
2015, Wiese et al. 2016).  The effect of glacial isostatic adjustment is removed from GRACE(-FO)
products using ICE-6G_D model estimates (Peltier et al., 2017). The geocentre motion (degree 1)
coefficient is using the technique of Sun et al. (2016) (Technical Note 13).  Values of C20 (Earth's
oblateness) and C30 (for months after Aug 2016) are substituted with SLR data (Loomis et al., 2019). We
calculate solid Earth's elastic response by using the loading Love number of the Preliminary Reference
Earth Model (Wang et al.; 2012).
Estimates of GPS positions in ITRF2014 (Altamimi et al. 2016) are relative to center of mass (CM) in the
long term but relative to center of figure (CF) in the seasons (because ITRF2014 does not allow there to
be seasonal oscillations of CM).  We therefore remove the long-term rate of CM relative to CF to
transform the GRACE estimates in the long term from CF to CM (but do not remove seasonal oscillations
of CM relative to CF so as to preserve the ITRF seasonal frame relative to CF). The annual signal of the
geocenter (as realized by ITRF 2014) projected on the up component in north America on average
explains 3% of the GPS vertical displacement signal and can explain up to 20% for certain sites.
GRACE(-FO) vertical displacement monthly estimates are derived as follows (e.g., Davis et al., 2004):

$$U(\phi, \lambda) = a \sum_{l,m} \left( \frac{h_l^E}{1 + k_l^E} \right) P_{lm}(sin\lambda) \times [C_{lm}cosm\phi + S_{lm}sinm\phi] \qquad (1)$$

Where, $U$ is the estimate of vertical displacement, $a$ denotes the Earth's radius, $\phi, \lambda$ denote the latitude
and longitude, respectively; $P_{lm}$ are the associated Legendre polynomials, $k_l^E$ and $h_l^E$ are the elastic
gravity and vertical load Love numbers (Wang et al., 2012), respectively, and $C$ and $S$ are the spherical
harmonic coefficients derived from GRACE(-FO) monthly solutions with respect to degree $l$ and order
$m$. JPL releases gridded mascon fields, to derive spherical harmonics ($C$ and $S$ in Eq. 1). We transform
fields of equivalent water height to normalized harmonic coefficients using the inverse of Eq. 9 in Wahr
et al. (1998).  Like GPS, we subtract the GRACE(-FO) vertical displacement field of September 2012
from each monthly field to establish a common reference basis. GRACE(-FO) fields are estimated at a
0.5-degree spatial resolution ($\phi, \lambda$ in Eq.1). Thus, we extract GRACE(-FO) estimates at the station level
by interpolating bilinearly the vertical displacement from the nearest 0.5-degree grid point neighbors to
the station's location.
2.3 Screening metrics
GPS vertical displacement estimates are evaluated against the ones derived from GRACE(-FO), to assist
in identifying outliers or further corrections that may be needed. We employ a number of different metrics
to evaluate the agreement between the two datasets, and to determine whether to include it in the joint
solution or not. Similar to Yin et al. (2020) we quantify correlation and variance reduction between GPS
and GRACE(-FO) vertical displacements. The structure of surface mass periodic signals (e.g., annual
cycles, trends) as picked up by the two measurement techniques, also entails critical information
regarding mismodelled offsets, and is evaluated as well.
This process flags sites that need correction and corroborates joint inversion's hypothesis (Argus et al.,
2021), that a basic level of agreement is needed for the GPS data to be used to infer surface mass change.

*Correlation*
First, we specify the level of agreement between the datasets by estimating the Pearson correlation
coefficient between GPS and GRACE(-FO) timeseries. On average correlation is 62%, but stations
located on the West coast exhibit an agreement higher than 80%, which in most cases is driven by the
larger annual signal amplitude there. A more detailed look into the correlation metric is performed to
evaluate the agreement of GPS/GRACE(-FO) in retrieving the seasonal cycle amplitude in different
watersheds. We fit and remove a trajectory model $y(t)$:

$$y(t) = a + bt + A sin(2\pi t) + B cos(2\pi t), \qquad (2)$$

with $a$ being the intercept; $b$ being the trend and $A$ and $B$ being the amplitudes of the sine and cosine
components of a periodic function. In a future release of the dataset, we will evaluate the presence of
draconitic periods in the time-series and add them in the trajectory model if justified. With the timespan
of the current timeseries being up to 15 years, we cannot resolve for the draconitics (i.e., the first
draconitic period (351.6 days) and the annual cycle (365.25 days) are very close and require a long time-
series to be deciphered). For a more thorough discussion we refer the interested reader to Amiri-Simkooei
et al. (2017) and Klos et al. (2023).
We classify stations in watersheds and plot the GPS-GRACE(-FO) correlation coefficient (R) of each
station in different watershed against the amplitude of annual signals (Fig. 2b). To quantify the
relationship between magnitude of the annual cycle and correlation between the two datasets we fit a
linear function between the magnitude of the annual signals and the GPS-GRACE(-FO) vertical
displacement correlations for each watershed, separately. A steep slope (**$a$**) of the fit (**$a$**>0.5) indicates an
agreement between the two datasets, which depends on the magnitude of the annual cycle. This
relationship breaks when stations of a basin exhibit smaller annual cycles. We discuss an interesting case
in Supplements, where stations located in the Great Lakes region (part of the St. Lawrence watershed)
demonstrate a negative trend **$a$** $= -1.26$. The disagreement is even more pronounced while assessing the
second metric (i.e., trends). Both metrics, when taken together, helped us identify the source problem (i.e.,
unlogged offset that affected nearly 25% of the stations located in the St. Lawrence watershed) and take
corrective actions (see Supplements for more details). Note that for Figs. 2 and 3 the corrected data were
used.

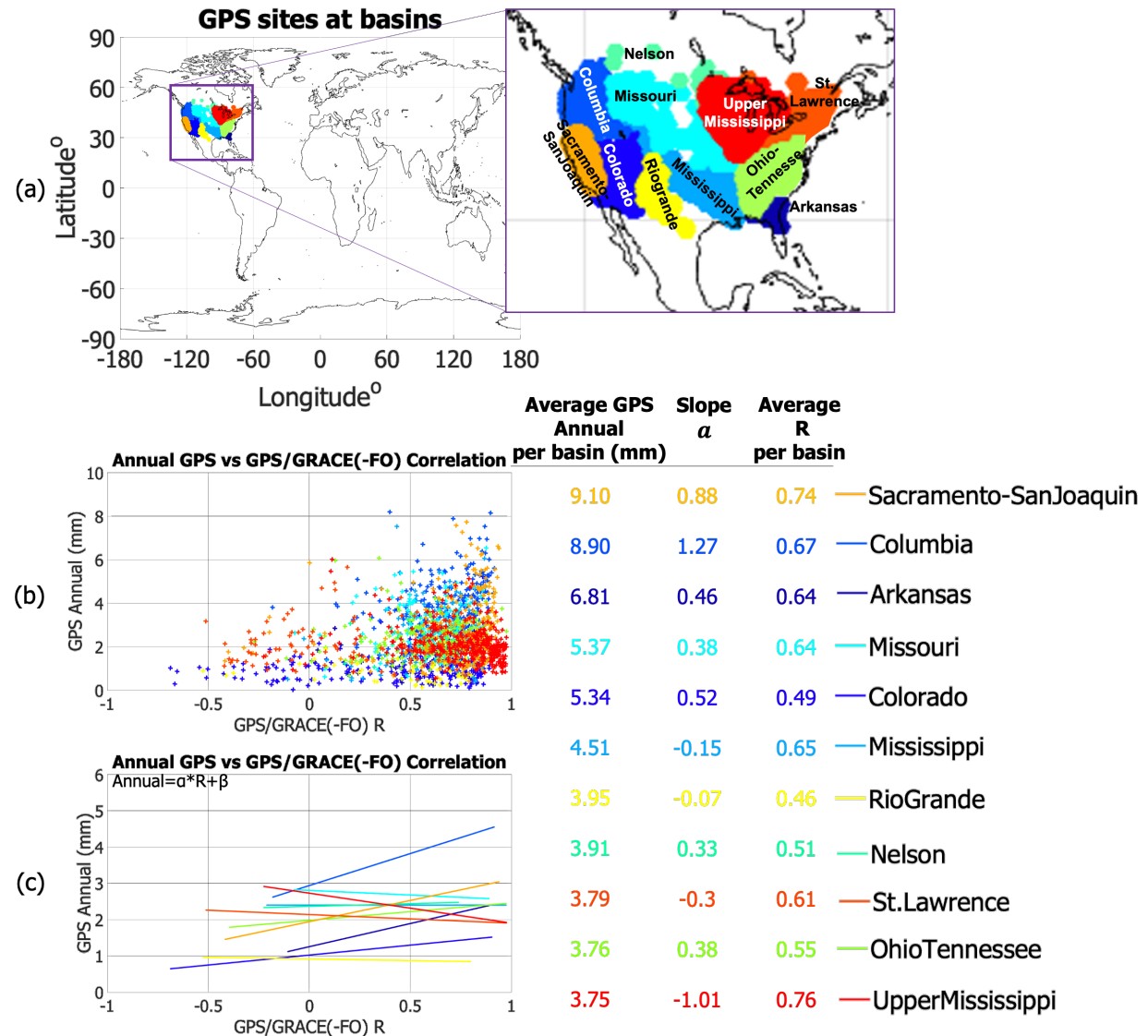

Figure 2: a) GPS sites clusters at watersheds in the US. Each watershed has a different color; b) Magnitude of annual GPS vertical displacement cycles derived with respect to GPS-GRACE(-FO) correlation; c) Linear fit between magnitude of the annual GPS vertical displacement cycles and GPS-GRACE(-FO) correlation.

*Trends*

In order to study the agreement between GPS/GRACE(-FO) in more detail, we split the timeseries of each station into non-overlapping intervals of 36 months, and fit Eq.2 for each station during each time-window. Different time-lengths of the GPS series may lead to misinterpretation of the geophysical content. For example, a station that has records only for the first 13 months out of the total of 36 months window may reflect different fit constituents compared to a neighbor station with full records, if the actual behavior of Earth's response changes during the 36-months window. Although in our dataset this

case is rare, we proceed with deriving the rate (slope) and the annual cycles only for stations that have records for at least 28 out of the 36 months. We did not interpolate the series during the GRACE(-FO) gap; thus, the last time-window reflects trends estimated using only GRACE-FO and GPS timeseries between June 2018-2021. As expected, GPS rates feature higher spatial variability than GRACE(-FO). However, both techniques capture large-scale quasi-periodic variations every 3 years (Fig. 3), an agreement that is noteworthy. The effect of this metric to detect outliers is pronounced when the two techniques show flipped trends.

Regions with pronounced trend disagreement:
- St. Lawrence watershed (stations located in the Great Lakes region at the State of Michigan): The trend during 2015-2018 was flipped between GPS and GRACE(-FO) in 62 stations (St. Lawrence watershed has a total of 243 stations available between 2015-2018). We discovered a missed offset in the series occurring in April 2016, and corrected for it, which led to an improved agreement in the trend (see Supplements).
- Cascadia region (northwest coast): The disagreement is evident in maps spanning 2009-2012, 2015-2018 and 2018-2021.5. GPS sites record a large surface uplift, which over the course of 15 years sums to 60 mm in sites located in Vancouver Island. GRACE(-FO) does not capture any such behavior. We attribute this disagreement partly on 1) glacial isostatic adjustment modeling error which manifests oppositely on two techniques. ICE6G_D predicts too much subsidence, thus when we correct GPS, we find too much uplift and when we correct GRACE(-FO) we find too much water gain which predicts too much subsidence; and partly on 2) the interseismic strain accumulation correction applied in the GPS dataset over this area (Argus et al., 2021). The sites have been flagged and are not going to be used in the joint inversion.
- San Andreas Fault (Southern California): Sites located in a vicinity of the Parkfield segment of the fault (Carrizon plain), exhibit consistent disagreement in the trend. More investigation is required to understand the mechanism that the fault presents on GPS/GRACE(-FO) vertical displacement estimates. The disagreement is also seen in Argus et al. (2022, Fig. S12). The sites have been flagged and are not going to be used in the joint inversion.

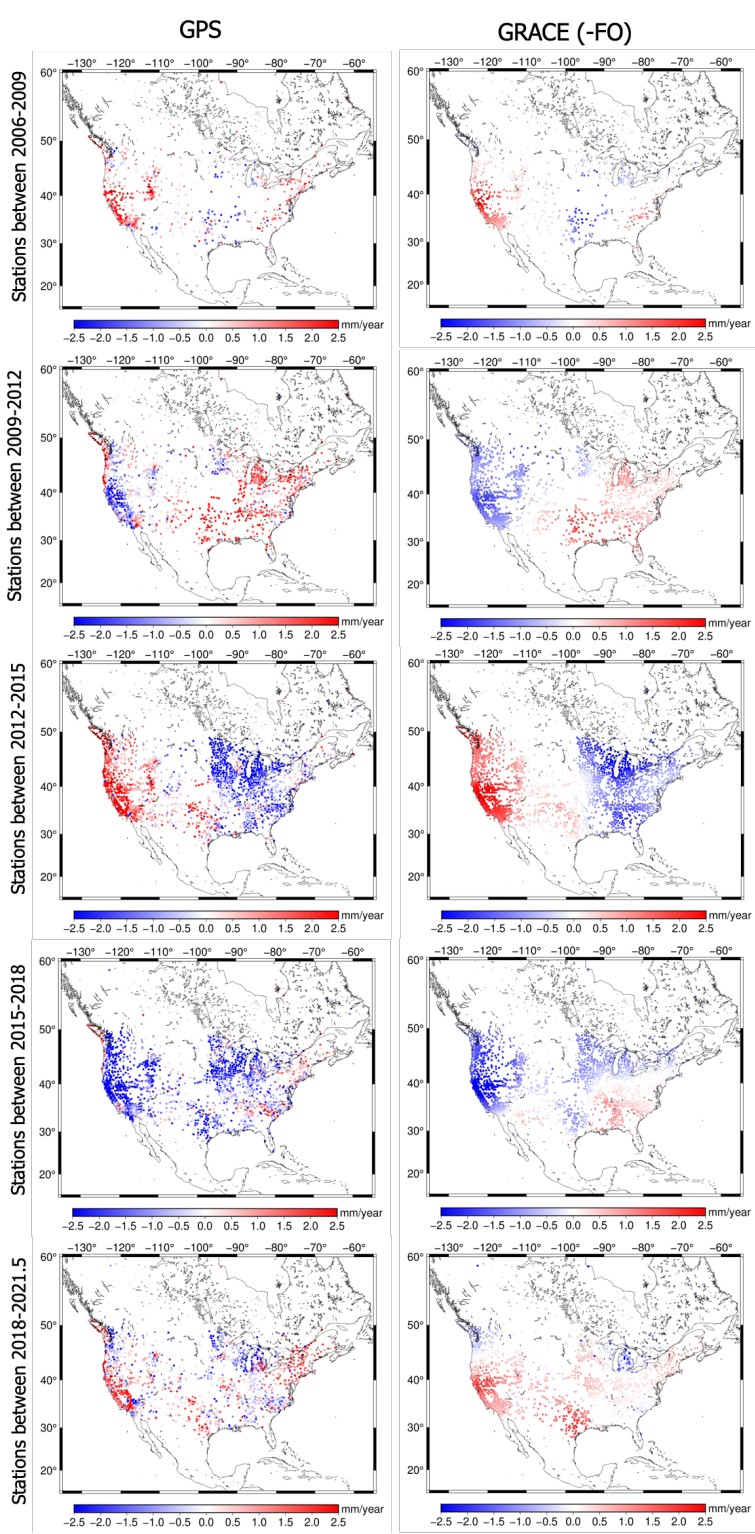

Figure 3: Rates of vertical displacements derived by GPS and GRACE. The rates are calculated every 36-
months (3 years) between 2006-2021.

*Variance Reduction*
Similarity in both amplitude and phase between two quantities is quantified via the variance attenuation
factor (Gaspar and Wunsch, 1989; Fukumori et al., 2015):

$$var_{red} = \left( 1 - \frac{var(GPS - GRACE(-FO))}{var(GPS)} \right) \times 100 \qquad (3)$$

The higher the agreement in phase and amplitude between GPS and GRACE(-FO), the closer the metric
gets to 100%. $var_{red}$ may also be negative when the differences in amplitude and/or phase are large.
Overall, GPS and GRACE(-FO) are consistent when $var_{red}$ exceeds 50%. The areas of main
disagreement are near coasts, especially along the Atlantic Ocean. This inconsistency can be partly
explained by modeling errors of the non-tidal oceanic and atmospheric loading model (e.g., Klos et al.,
2021; van Dam et al., 2007). Additionally, agreement is poor for sites located in the vicinity of the
Parkfield segment (specific regions across the fault perform poorly), which is consistent with the
disagreement shown in Fig. 3.

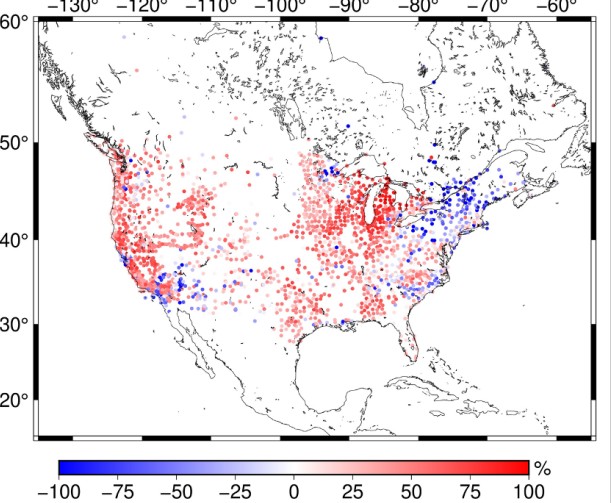


Figure 4: Variance reduction between GPS and GRACE(-FO) vertical displacements
We also compared the annual amplitudes of GPS and GRACE(-FO) vertical displacements (cosine and
sine components in Eq. 2). This analysis was not informative for the presence of outliers or errors in the
current data sample studied.
Overall, the screening process not only assisted in outlier detection, but it also allowed for a deeper look
into the structure of vertical displacement periodic signals. We identified the need for antenna offset
corrections (in sites located in the Great Lakes region); removed sites affected by glacial isostatic
adjustment and interseismic modeling errors; and sites located at the Parkfield segment of San Andreas
Fault.

## 3. Uncertainty Quantification

With the updated dataset we are now ready to proceed with the uncertainty quantification of the GPS vertical displacement timeseries. We apply different error characterization schemes consisting of a root sum square of a random error, white noise error, power law noise error (flicker noise and random walk) and spatially coherent error.

3.1 Methods

*Root Mean Square Error*

Residuals $r$ of a series with respect to a trajectory model (Eq. 2) are often used as a first approximation of noise in vertical displacement series (e.g., Bos et al., 2013; Michel et al., 2021). Practically, $r$ shows how well a trajectory model can describe the original timeseries. Therefore, the root mean square (rms) of $r$ can give a first approximation of the noise floor of each station.

*Spectral Analysis, White, Flicker and Random Walk Noise*

Power distribution of residuals and its agreement with noise models, is another popular way to quantify uncertainty of GPS timeseries (e.g., Klos et al., 2019; Argus et al., 2022). Typically, GPS series are evaluated for white, flicker and random walk noise, or combination of them. Hector software (Bos et al., 2013) is used to estimate full noise covariance information by means of a maximum likelihood estimator. The covariance matrix $C$ from a combination of white and power law (i.e., flicker and random walk) noise is given as:

$$C = a \times I + b \times J \hspace{3cm} \text{Eq. 4}$$

Where $a$ is the amplitude of white noise, $I$ is the identity matrix of size N (number of samples/epochs in the series), $b$ is the amplitude and $J$ the covariance matrix of power law noise. $J$ matrix is a full covariance matrix that describes the time-correlated error (as the data record length increases, the displacement uncertainty changes (Bos et al., 2008 Eqs. 8-11)). The optimal selection of the noise models is done via two optimality criteria, namely the Akaike Information Criterion (Akaike, 1974) and the Bayesian Criterion (Schwarz, 1978).

In this study, we consider three cases:
a)      White Noise (WN)
b)      Combination of WN and Flicker Noise (WN+FN)
c)      Combination of WN, FN and Random Walk Noise (WN+FN+RW)
We take the root-sum-squares of the noise magnitudes as our noise floor. For example, for the case of WN+FN noise, noise is derived as $\sigma = \pm\sqrt{\sigma_{WN}^2 + \sigma_{FN}^2}$. Our data are sampled on a monthly basis, thus

$\sigma_{FN}$ needs to be scaled appropriately, i.e., $\sigma_{FN} = \sigma_{PL}(\frac{1}{12})^{-\frac{k}{4}}$, where, $\sigma_{PL}$ is the uncertainty of power-law
(PL) and $k$ the spectral index, outputted from Hector (more information on power-law noise estimation
can be found in Bos et al., 2008, and Williams, 2003).

*Common Mode Noise*

The Common Mode Component (CMC) is derived following the processing scheme suggested by
Kreemer and Blewitt (2021), which can be summarized as:

1) Input GPS displacement timeseries (referenced to Sep 2012) for $j$ stations ($l_j$)
2) Derive each station's residuals by removing the trajectory part of the series ($l_j(t) - y_j(t)$)
3) Quantify the correlation coefficient $r_{MAD}$ using robust statistics. $r_{MAD}$ is defined as:

$$r_{MAD} = \frac{MAD^2(u) - MAD^2(v)}{MAD^2(u) + MAD^2(v)} \qquad \text{Eq. 5}$$


The median absolute deviation ($MAD$) is the absolute deviation around the median. For example, for a
residual series res(t) $MAD = |res(t) - median(res(t))$. $u$ and $v$ are derived as:

$$u = \frac{p - median(p)}{\sqrt{2}MAD(p)} + \frac{q - median(q)}{\sqrt{2}MAD(q)} \qquad \text{Eq. 6}$$

$$v = \frac{p - median(p)}{\sqrt{2}MAD(p)} - \frac{q - median(q)}{\sqrt{2}MAD(q)} \qquad \text{Eq. 7}$$


with $p$ and $q$ being the residual series of the reference station and the neighbor station, respectively.
For each station there are $j - 1$ correlation coefficients $r_{MAD}$. In order to decide the cut-off distance
that a neighbor station will be considered in the analysis we plot $r_{MAD}$ coefficient against its distance
from the reference station (Fig. 5). Based on results from all stations we decide to set a cut-off at 1500
km, slightly higher than the 1350 km suggested by Kreemer and Blewitt (2021). The 1500 km cut-off
allows us to separate stations between East and West coast, as spatially coherent signals at stations
located across the continent are negligible.
4) Derive the median slope estimator ($ccs$) using Theil-Sen median trend. $ccs$ is the median trend of the
$r_{MAD}$ coefficients of a station against their distance with the reference station.
5) Derive the zero-distance intercept $cci_j$ for each station as median($r_{MAD} - ccs * d$), with $d$ being the
distance between the station of reference and the neighbor station (maximum $d$ = 1500 km).
6)  Construct CMC: Calculate the cumulative ($c_j$) and percentile ($p_j$) weights for each station and then
422       find the weighted median that corresponds to $p_j = 50\%$. This weighted median represents the CMC of
423       the station (Fig. 6).

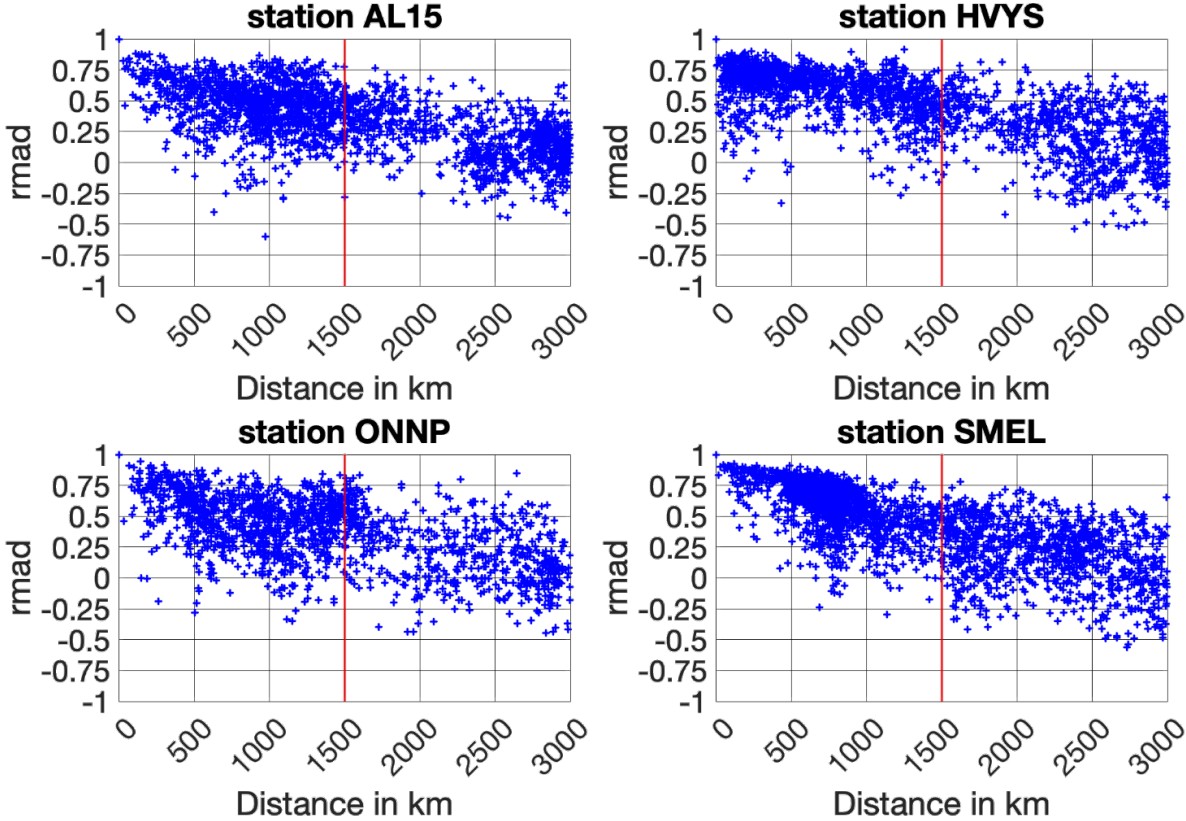

Figure 5: $r_{MAD}$ coefficient of four random stations with the rest of the station sample, plotted against the
distance of the reference station with the rest of the stations. Each cross resembles the $r_{MAD}$ of the
reference station with a station located at distance $d$.
CMC is limited in providing a realistic error approximation, in that the technique cannot isolate spatially
correlated noise from signal (e.g., hydrology signals not described by the trajectory model are present in
the residuals fed into CMC). Under the realistic assumption that a component of the high frequency signal
contained in CMC reflects real hydrological processes, we remove the contribution of surface hydrology
using Global Land Data Assimilation System (GLDAS) (Rodell et al., 2004) vertical displacement
estimates. GLDAS does not model deep groundwater and open surface water, so these signals remain in
the residual (Scanlon et al., 2018). Vertical displacement estimates driven by surface hydrology are
derived similar to GRACE(-FO) (Section 2.2). We use Noah v2.1 monthly estimates of soil moisture
storage given at 0.25-degree grids (Beaudoing and Rodell, 2016), convert the fields from terrestrial water
storage ($kg/m^2$) to units of equivalent water height, derive the spherical harmonic coefficients of the
equivalent water height mass load using Wahr et al. (1998), and predict the elastic response of the Earth
(Eq. 1). Afterwards, we remove the reference epoch (09/2012) similar to GPS and estimate the vertical
displacement at the locations of the GPS sites by interpolating the estimates of the closest neighbors to the
station's location.  Note, that because our interest is to prepare the data for a combined solution with
GRACE(-FO) we interpolate the timeseries at the times of GRACE(-FO) monthly series availability. The
interested reader is referred to the supplement, where we show the vertical displacement estimated by
GPS, GRACE(-FO) and GLDAS (Figure S2) for randomly selected stations. Finally, we derive residuals
relative to the trajectory model (Eq. 2). GLDAS (surface hydrology) residuals should ideally reflect high
frequency hydrological processes and are therefore removed from GPS residuals. Overall, CMC of
surface hydrology residuals exhibits a fairly small magnitude (~0.5 mm). We remove the contribution of
surface hydrology within the CMC algorithm by first subtracting GLDAS vertical displacement estimates
from GPS, and next inputting the residuals of this difference into the algorithm. The output of this process
($CMC_{HF}$) slightly decreases the magnitude of CMC and expresses a more realistic representation of
spatially correlated noise.
3.2 Results
Vertical displacement uncertainty of each station is estimated by means of all the different approaches
discussed in Section 3. Mean ($\mu$), median and standard deviation (std) values are shown in Table 1. On
average, an assumption of white noise shows slightly reduced uncertainty compared to the other
techniques, followed by RMSE. When flicker noise is considered in addition to white noise (WN+FN) the
average uncertainty increases by nearly 0.8 mm compared to the white noise only. We note that the
contribution of white noise in the case of WN+FN is negligible for ninety seven percent of the stations
(that is flicker noise describes the noise exclusively). Noise level from combination of all three noise
models (WN+FN+RW) is less than 4 mm on average. In this case too, white noise is negligible, and noise
is described exclusively from flicker noise for 1550 stations, and from random walk for 600 stations. The
rest of the data sample reflects a contribution from both noise models. We additionally analyzed the
amplitude of the noise of each noise model ($\sigma_{PL}$) with respect to the length of the input series. Results did
not identify any clear relationship between $\sigma_{PL}$ and the length of each station's timeseries.  CMC noise
floor is 3.6 mm on average with a relatively large standard deviation ($\pm 1.6$ mm) which suggests that
spatially correlated noise has higher variability than time-correlated noise ($\pm$ 1.6 mm as opposed to $\sim \pm 1$
mm). When surface hydrology is removed ($CMC_{HF}$) the noise floor drops by a fraction of a mm on
average compared to CMC.
Table 1: Different uncertainty quantification cases

|            | mean ($\mu$) (mm) | median (mm) | $\pm$ std (mm) |
|------------|-------------------|-------------|----------------|
| RMSE       | 2.8               | 2.7         | 0.8            |
| WN         | 2.4               | 2.2         | 0.8            |
| WN+FN      | 3.2               | 3.1         | 0.7            |
| WN+FN+RW   | 3.8               | 3.5         | 1.1            |
| CMC        | 3.6               | 3.2         | 1.6            |
| $CMC_{HF}$ | 3.5               | 3.1         | 1.6            |

RMSE and WN exhibit a smooth transition among the regions, which indicates the presence of spatially
coherent regime signal mostly driven by hydrology (Fig. 6). The combination of WN+FN is mostly
dominated by FN and the uncertainty exhibits local (in space) coherence. The uncertainty is larger when
random walk is included in the combination (WN+FN+RW). A recent study from Argus et al. (2022) on
groundwater flux in Central Valley (California) suggests that noise on GPS-derived uplift motion can be
well described by a combination of flicker noise and random walk, due to the ability of these noise
models to reflect low frequency noise. When a simulated contribution of the surface hydrological
component is removed from the series, $CMC_{HF}$ reflects a more realistic picture of the noise. Arguably the
level of change compared to CMC is sub-millimeter. Signal contributions from un-modelled groundwater
variations are potentially still present, but groundwater changes are typically slower in time.
We obtain the relative likelihood of each uncertainty quantification method by estimating the probability
density function (PDF) (Fig. 7).  White noise has a flat power spectrum, having the same amplitude
across frequencies. Estimating a best fit for a flat spectrum doesn't allow for capturing the long tail skew
of the residuals (low frequency), which are biased towards their mean. Thus, the amplitude of white noise
is smaller compared to the rest of the techniques (Table 1). Flicker and random walk noise models add to
the long tail of the power distribution, that is they allow more low frequency noise, which explains the
higher amplitude of the uncertainty when these two noise types are considered.
RMSE and WN show a 50% probability of a station having an uncertainty ($\sigma$) between 1.5-2 mm and less
than 10% of a station exceeding $\sigma$=4 mm. The noise level fells within [2 4] mm for ~93% of the stations
when we consider combination of WN+FN. PDF of RMSE, WN and WN+FN resemble a normal
distribution, with the mean being shifted for each case. When random walk is also considered
(WN+FN+RW) 64% of the stations exhibit noise within [2 4] mm. In this case, the distribution is more
spread resembling a gamma-like distribution, with a peak being at 3 mm (18%). CMC and $CMC_{HF}$ PDF
also follow a gamma-shape, and the probability of the uncertainty ranging between [2 4] mm is nearly
60% for CMC and 65% when surface hydrology is removed.

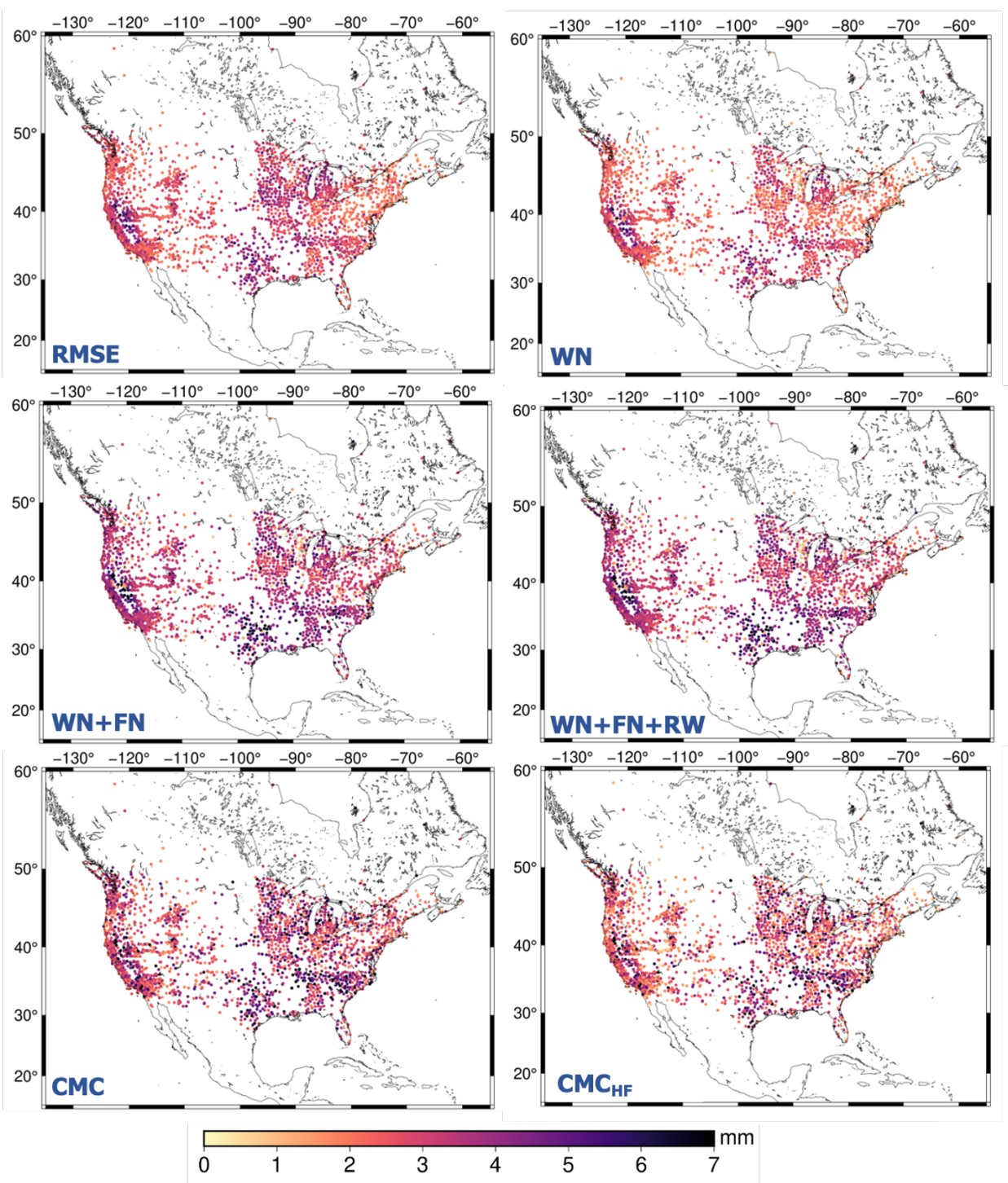

Figure 6: Noise amplitudes of GPS timeseries estimated using different techniques.

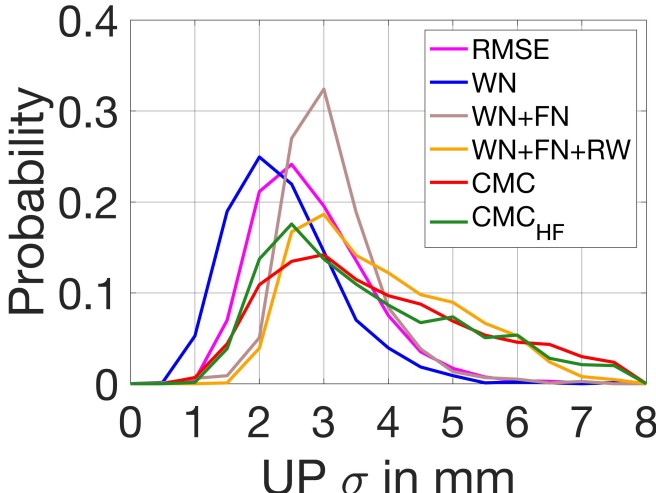

Figure 7: Probability density function of vertical displacement estimates uncertainty

## 4. Conclusions and Discussion

GPS-derived vertical displacements are very useful for supplementing GRACE(-FO) gravity products to infer mass change signals at spatial scales smaller than what can typically be achieved with current satellite gravimetry alone (i.e., < 300km). This work provides a general workflow to isolate elastic surface mass signals from GPS vertical displacement, by developing processing standards; additionally, it suggests uncertainty quantification schemes to quantify error on GPS vertical displacement estimates. The ultimate goal is to prepare GPS estimates for merging with satellite-gravimetry observations. First, we provide a list of corrections needed for isolating surface mass following recommendations outlined in Argus et al. (2017; 2022). Additionally, a detailed investigation of trends, correlation, and variance reduction highlights the need for better background modeling (glacial isostatic adjustment and interseismic strain), as the two observation techniques respond differently in the presence of such errors. At this point the recommendation is to remove sites located in the vicinity of regions where background models are known to perform poorly, before any joint inversion. Except detecting outlier stations, screening metrics point to extra corrections that need to be applied in certain sites (e.g., missed antenna offsets).

Several uncertainty quantification schemes have been tested to prescribe weights on GPS vertical displacement estimates that are needed for a joint inversion with GRACE(-FO) data. The average noise level indicated by RMSE is 2.8 mm. White noise average is 2.5 mm. The errors increase when lower frequencies are included in the noise estimation. When we account for flicker noise, one third of the sites exhibits noise levels of up to 3 mm. The average noise increases significantly in presence of random walk, as more power of the lower frequencies gets into the estimations, and the distribution of noise is more dispersed. In this case, half of the stations are prescribed with > 4 mm uncertainty. Argus et al. (2022), finds that random walk is the most realistic representation of noise based on postfit residuals. We notice that the spectrum of CMC provides similar uncertainties to random walk, which implies that despite the different characterization procedure, CMC is able to provide equally realistic noise estimates of GPS timeseries. We attempted to minimize lingering hydrology signals embedded in CMC, through

reducing the GPS vertical displacement observations with displacements from the GLDAS hydrology
model. The average noise floor dropped slightly (~0.5 mm drop in sigma). Future work will provide
further information of GPS station errors when the weight of each GPS site is also considered based on its
impact on the performance in a formal data combination of GPS-GRACE(-FO). The suggested
framework can be easily adjusted to account for global datasets. The new dataset provides GPS vertical
displacements of elastic mass variations in North America and their associated uncertainties.

**Data Availability:** The data product described in the manuscript is available in zenodo (doi:
https://zenodo.org/record/8184285). GPS timeseries are provided by the Global Station List from the
Nevada Geodetic Laboratory (http://geodesy.unr.edu/; Blewitt et al., 2018). Non atmospheric and oceanic
tidal aliasing product (AOD1B RL06) is provided by GFZ's Information System and Data Center
(ftp://isdc.gfz-potsdam.de/grace/Level-1B/GFZ/AOD/RL06, Dobslaw et al., 2017). GRACE and
GRACE-FO Level 2 products are available from podaac (https://doi.org/10.5067/GFL20-MJ060).

**Acknowledgments:** The research was carried out at the Jet Propulsion Laboratory, California Institute
of Technology, under a contract with the National Aeronautics and Space Administration
(80NM0018D0004). Maps were made with the Generic Mapping Toolbox (Wessel et al. 2019). We thank
Corne Kreemer (UNR) for his feedback and Mike Heflin (JPL) for his insights on draconitic errors.

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
