# Peer review of "GPS displacement dataset for study of elastic surface mass variations"

_Earth System Science Data, 2023_

## Author Comment (AC1)

################################## Reviewer 1 ##################################

This paper presents some procedures that help in making vertical land displacements useful as complementary data to GRACE for inverting hydrological loads, including noise analysis, using discrepancies between GPS and GRACE to detect missed GPS offsets, and an improvement of the CMC Imaging method of Kreemer and Blewitt (2021). The data produced is ultimately very useful for researchers interested in hydrologic loading. Where will the VLD time-series be available?

Thank you for your comment. We provide the data repository in the Data Availability Section: "Data Availability: The data product described in the manuscript is available in zenodo (https://doi.org/10.5281/zenodo.8184285)."

Possible issue with Correlation presentation:

The calculation of correlation between GPS and GRACE (per watershed) is not clear to me, given that you have X GPS time-series and Y GRACE mascon time-series. I suspect there is a missing step, that doesn't seem to be explained: how GRACE data is translated to VLD at the station level. Please explain that.

Thank you for your comment.

0. We derive GRACE(-FO) VLD fields (lon, lat, Up displacement) using Eq. 1 (described in Davis et al., 2004) for each month that GRACE(-FO) was available starting in 2006 (because our GPS timeseries begin in 2006).

1. Subtract the reference period (09/2012) from each monthly up field.

2. GRACE(-FO) fields are estimated at a 0.5-degree spatial resolution (Eq.1). Thus, we extract GRACE(-FO) estimates at the station level by interpolating the vertical displacement from the nearest 0.5-degree grid point neighbors to the station's location.

We added the following sentences in the main text.
JPL releases gridded mascon fields, to derive spherical harmonics ($C$ and $S$ in Eq. 1). We transform fields of equivalent water height to normalized harmonic coefficients using the inverse of Eq. 9 in Wahr et al. (1998).  Like GPS, we subtract the GRACE(-FO) vertical displacement field of September 2012 from each monthly field to establish a common reference basis. GRACE(-FO) fields are estimated at a 0.5-degree spatial resolution ($\phi, \lambda$ in Eq.1). Thus, we extract GRACE(-FO) estimates at the station level by interpolating bilinearly the vertical displacement from the nearest 0.5-degree grid point neighbors to the station's location."

Furthermore, I don't see how a poor correlation can be attributed to a single station (as the case for St. Lawrence) ; this is assuming that the authors didn't mean to say that there was a missed GPS offset in all of the time-series, which seems unlikely.

You are right. The issue with the missed offset affected almost 25% of the stations located in the St. Lawrence watershed. We revised the manuscript and provide the exact number of stations affected (62) relative to the total number of sites in the watershed (243) (See Line 272).

Minor points:

Line 85. Kreemer and Blewitt did not introduce the term CMN, but rather used the term CMC (common mode components), which was first introduced by Tian and Shen (2016)

True. Thank you. We changed the reference to Tian and Shen, 2016.

Line 118. Originally is not used properly in this contextLine 118. "layout" should read "lay out"

Thank you. Done!

line 126-127: "We overcome CMC's limitation of include spatially correlated hydrology signals...". This sentence is grammatically not correct, nor is the context clear: which limitation? That includes both noise and unmodeled signal?

We strive to isolate noise from signal, thus we were interested in deriving the common-mode component reflecting surface hydrology signals and removing it from the respective GPS timeseries. We revised the sentence accordingly: We build on the existing CMC algorithm to remove hydrology signals from the error estimate by deriving surface loading signals from a hydrology model and removing them from the GPS up displacements (see Section *3 for more details)*.

Line 192: I think P_lm is missing in front of "are the associated Legendre polynomials"

Thank you. The typo is now fixed.

Line 220: "*A* and *B* being the amplitude and phase". That is not how formula 2 is written, which is erroneous, should be A*cos(2*pi*t+B)

Thank you. The typo is now fixed.

Line 302-303: "We identified the need for antenna offset corrections (in the case of Great Lakes)". Before the St. Lawrence watershed was named in this context, now Great Lakes. Later (line 458) Lake Michigan is mentioned. Which is it?

Thank you. The sites are located in the Great Lakes region of the State of Michigan, which belongs to the southern west end of the St. Lawrence watershed. To avoid confusion, we refer consistently to the sites, as "sites located in the Great Lakes region" and we provide some more detail on the number of the stations affected by the offset issue.

---

## Author Comment (AC2)

The article is a valuable compilation of methods and approaches used in the geodetic literature to analyze time series. However, it contains several shortcomings that limit its understanding or use of the dataset. The main limitation is that the authors do not make the created dataset and its uncertainties available (as they promised in the abstract and introduction), which greatly prevents me from assessing its potential. Some of the results are presented laconically (as the uncertainties; I presume these are the uncertainties of trend, or…?), and others lack adequate explanations (where is the RW in the data coming from? Perhaps it is the result of inadequate series length? or other effects?). The authors are keen to streamline the procedure for qualifying the data for further analysis, but sometimes speeding up preprocessing or classification can lead to erroneous conclusions.

Thank you for your comments and recommendations. Please find our response below:

Major shortcomings include:

1. I believe that a key requirement for publishing an article in ESSD is that the proposed dataset be available. Unfortunately, I found nothing about this either in the article or in the supplementary materials. Do the authors plan to make available GPS VLDs that can be taken directly for comparison? Their uncertainties were also described, but are not available. Will these be the uncertainties of individual observations or the uncertainties of trends?

   Thank you for your comment. Please refer to the Data Availability section where the doi of the GPS VLD timeseries and their uncertainties are made publicly available:

   "Data Availability: The data product described in the manuscript is available in zenodo (doi:10.5281/zenodo.8184285)." https://doi.org/10.5281/zenodo.8184285.

   The uncertainties reported in this study reflect the noise of the GPS VLD timeseries, not the trends. We clarified this in the manuscript.

2. It would also be useful to make available a list of stations that the authors considered being those responding elastically to the applied load, their coordinates, and a list of stations excluded from comparisons as those responding poroelastically.

   Thank you for the suggestion. When a station is marked as "not responding elastically", it does not necessarily mean it includes a porous response. Other reasons such as tectonic motion may explain the non-elastic behavior of the station. The root issue underlying a stations non-elastic behavior is not studied in this manuscript. The purpose of this study is to provide GPS timeseries characterized by an elastic solid Earth motion. Therefore, to avoid confusion we'd rather include only the product with the GPS stations that respond elastically.

3. The authors do not explain all the abbreviations used in the article.

Thank you for your comment. We revised the manuscript and explained the remaining abbreviations.

4. Please show some of the GRACE and GLDAS time series against the GPS time series so that readers can get a general idea of how the time series agree with each other. Not all users of the dataset need to be geodesists.

Thank you for your comment! We added a plot with GPS, GLDAS and GRACE(-FO) up estimates) in the supplements (S2) and we now explain in detail how we derived GLDAS up displacements in the manuscript (Section 3.1 lines 430-446).

5. It is not clear how the authors converted the TWS available for JPL mascons to displacements. Equation 1 does not describe the entire procedure.

Thank you for your comment. We revised the description of our GRACE-FO processing to be more specific. We also added the following sentence that cites the reference we used to derive $C$ and $S$:

"JPL releases gridded mascon fields, to derive spherical harmonics ($C$ and $S$ in Eq. 1). We transform fields of equivalent water height to normalized harmonic coefficients using the inverse of Eq. 9 in Wahr et al. (1998)."

Wahr, J., Molenaar, M. and Bryan, F., 1998. Time variability of the Earth's gravity field: Hydrological and oceanic effects and their possible detection using GRACE. *Journal of Geophysical Research: Solid Earth*, *103*(B12), pp.30205-30229.

6. It is also unclear how the authors interpolated the gap between GRACE and GRACE-FO. This is because they present trend estimates for the period 2018-2021.5, which includes a gap of more than a year in GRACE observations.

Thank you for the comment! You are correct about the missing data due to GRACE/GRACE-FO gap. As described in the introduction our main interest is to process GPS VLD estimates for use in GRACE/GRACE-FO combined solution. Thus, we do not interpolate for the missing GRACE(-FO) months.

The last time-window contains GRACE-FO data starting in June 2018 that extend until June 2021 (3-years). Therefore, trends reported for the final time-window are derived using GPS VLD estimates only during the epochs that GRACE-FO data are available. We revised the manuscript accordingly.

*We did not interpolate the series during the GRACE(-FO) gap; thus, the last time-window reflects trends estimated using only GRACE-FO and GPS timeseries between June 2018-2021.*

Equation 2 lacks time dependency y(t).

Thank you. Fixed.

7. It is not clear what "ccs" means in the CMN algorithm.

ccs is the slope of the $r_{MAD}$ coefficient with respect to the distance of the station to the reference site (Fig.5 shows the $r_{MAD}$ coefficient wrt the distance). Please see the referenced work of Kreemer and Blewitt (2021) about the CMC processing steps. To avoid confusion, we added in the main text:

*ccs* is the median trend of the $r_{MAD}$ coefficients of a station against their distance with the reference station.

8. It is not clear with which noise model the authors perform the analysis for CMC and CMCHF.

CMC and consequently CMC_HF provide a robust way to quantify spatial correlation of the noise. Please refer to the Paragraph *Common Mode Noise* in Section 3, that describes the way spatially correlated noise is estimated using CMC technique.

9. Table 1: What μ means here? Why the authors do not compare the amplitudes of noise or the percentage contribution of different noises to the analyzed combinations? This might be interpretable.

Thank you for your comment. μ signifies the mean. We added the description in Section 3.2. Mean (μ), median and standard deviation (std) values are shown in Table 1.

We also found your recommendation on quantifying how much each noise model contributes to the combined power law noise very constructive. Instead of actual numbers we report the percentage contribution of each of the noise models in the total noise.

There are two cases where noise models are combined as described in the manuscript, the case with white noise being combined with flicker noise (WN+FN) and the case with white, flicker and random walk (WN+FN+RW) noise being all combined together. In the first case, flicker noise dominates the spectrum in the vast majority of stations. In particular, white noise is almost zero for 2943 stations (describes >80% of the total noise for only 87 stations in our whole datasample).

We also analyzed the amplitude of the noise of each noise model with respect to the length of the series. No clear picture is drawn between this amplitude, ($\sigma_{PL}$ as described in the manuscript and on Bos et al., 2008;2013) and the length of the series. We added the following statement to the main manuscript:

"We additionally analyzed the amplitude of the noise of each noise model ($\sigma_{PL}$) with respect to the length of the input series. Results did not identify any clear relationship between $\sigma_{PL}$ and the length of each station's timeseries."

In the second case (WN+FN+RW) white noise is again almost entirely absent, and noise is driven by a combination of flicker and random walk. Therefore, we will only analyze the percentage contribution of FN and RW.

We plot a hexbi diagram to describe the relative contribution of each noise model to the total noise. The percentage contribution of each noise model is resembled by hexagonal bins, and the color of the bins represents the number of stations that exhibit this particular percentage. Histograms that reflect the number of stations having a certain percentage [0 1] are overlayed on the two axes. The relative contribution of each of the noise models is shown in the figure below. Half of the stations (~1550) exhibit exclusively flicker noise (dark green hexagon centered at 1 on FN axis), 600 stations exhibit exclusively random walk noise (hexagon centered at 1 in the RW axis), and the rest of the data sample (880) stations is partially described by both noise models.

[Figure]

We added a short discussion in the main manuscript and the supplement. Thanks again for the recommendation!

10. Figure 6: Is this the uncertainty of trend, or…

This is the uncertainty of the VLD timeseries and not the trend's uncertainty. To avoid confusion, we added the following description under Fig.6 label: *Uncertainty of GPS timeseries estimated using various techniques.*

The authors mentioned a case of unlogged offset. It is presented in the supplementary materials, but only for one case. I think the authors should approach the topic more descriptively and present other cases in which unlogged offsets were also corrected manually.

Thank you. The unlogged offset contaminated almost 25% of the stations located at the St. Lawrence watershed. Please see the revised version of the manuscript.

11. Lines 229-230: The authors mention describing interesting cases in Supplements, but they are missing there.

Please let us copy Line 263. "We discuss an interesting case in Supplements, where stations located in the St. Lawrence basin demonstrate a negative trend $a = -1.26$."

We provide more analysis in the case of St Lawrence in the Supplementary material. As mentioned above, a quarter of the stations of this watershed experienced this offset.

Several sentences in the text are missing a noun or verb, the authors should carefully review the entire text.

Thank you! We have reviewed the sentences.

---

## Author Response (AR2)

The paper still needs a few major corrections:

1. Lines 25-26: the wording of „each technique" and „three techniques" is not clear

Thank you. We revised the sentence.

2. Line 80: 'the reported uncertianty of the measurement'. I guess you are not reporting the uncertinaty of GPS measurements, but the uncertainty of the resulting displacements…?

You are right. We revised the sentence.

3. Line 115: The trajectory model has evolved over the years. The one you've mentioned here is the simplest trajectory model, which has been upgraded to an extended model best suited to displacement series.

The trajectory model has evolved with time. From linear only fit, to linear fit with jumps, to the most widely used fit (offset, rate, seasonal). Some more modern models such as the extended linear trajectory can also handle postseismic deformation.

Our data pre-processing handles the cases that could potentially need the model with postseismic transient (see Paragraph 2). For example, we delete data that may be biased by a postseismic transient, so no post-seismic fit is needed. Therefore, we prefer using the standard conventional trajectory model (see e.g., Klos et al., 2023).

4. Did the authors quantified the impact of draconitic period and thermal expansion of bedrock/monument on their annual signal they are comparing? This may be of large importance when comparing the correlation vs annual amplitude.

Draconitics:

Literature: Argument for there being a significant draconitic in GPS data biasing the interpretation of position-time series is underwhelming.

- As the reviewer suggests, the first draconitic (351.6 days) is very close to annual cycle (365.25 days). To resolve for the differences between the two a minimum of 15 years of data is required (Klos et al., 2023). Our timeseries do not exceed 15 years, thus we do not add the draconitic period to the determinist model.
- Several tens of refereed articles on GPS measurements of seasonal oscillations without there being mention of a draconitic [e.g. White et al. 2022]. We rely on the satellite orbit determination of Bertiger et al. 2002 and the site position determination in Blewitt et al. 2018; we do not want to alter these documented results.
- In the GPS contribution to ITRF 2022 [Altamimi et al. 2023], Rebischung 2022 estimated and removed periodic signals at the first 8 GPS dracontic harmonics [https://itrf.ign.fr/en/solutions/ITRF2020]. However, there are no specifics on how big the amplitude of the draconitics are (https://itrf.ign.fr/docs/solutions/itrf2020/IGS-contribution-to-ITRF2020.pdf ). Ray et al. 2008 is the seminal study on the GPS dracontic. But Amiri-Simkooei et al. 2017 find the draconitic in the 3rd GPS reprocessing to have decreased and to be minor.

To further assess, we fit an offset, a rate, a sinusoid with a period of 1 year, and a sinusoid with a (first dractonitic) period of 351.6 days to JPL's X files (the file contains transformation parameters and position and velocity residuals relative to ITRF2014), the values X, Y, Z, and Earth's scale applied to all positions. Overall, daily X files between 1992-2023 are used for the analysis.
For all 4 quantities, we find the amplitude of the first draconitic to be no more than 1-2 mm.

We appreciate the reviewer's feedback. We will keep in mind and evaluate further as we prepare a next manuscript where longer GPS timeseries will be available; right now, we want to keep with the GPS position-time series of Blewitt et al. 2018 (2006-present).

In the figure below:
light blue shows daily X,Y,Z and scale parameters with respect to ITRF2014.
Top 4 panels: A bias, rate and sinusoid fit is shown in dark blue. The rate and the amplitude of the peak-to-peak oscillation are reported.
Bottom 4 panels: A bias, rate, sinusoid and draconitic (351.6) fit is shown in dark blue. The time-series is long enough and allows for deciphering between annual cycles and draconitics. The amplitude of draconitic using 30+ years of data ranges between 1-2 mm (X and Z axis respectively).

[Figure]

We added the following short explanation in the manuscript:
*In a future release of the dataset, we will evaluate the presence of draconitic periods in the time-series and will add them in the trajectory mode if justified. With the timespan of the current time-series being up to 15 years, we cannot resolve for the draconitics (i.e., the first draconitic period (351.6 days) and the annual cycle (365.25 days) are very close and require a long time-series to be deciphered). For a more thorough discussion we refer the interested reader to Amiri-Simkooei et al. (2017) and Klos et al. (2023).*

Thermal expansion: Thermal expansion can show up in annual cycle. Typical values of thermal expansion suffice to rule out annual vertical signals driven heating and cooling of the bedrock (Tsai, 2011). For example, an $8 \times 10^{-6}$ °C$^{-1}$ linear coefficient of thermal expansion times 2 m depth times a 30 °C seasonal temperature variation delivers an estimate of motion of just 0.5 mm (Argus and Peltier, 2014). Klos (2023) finds that amplitude of thermal expansion is anywhere between 20-40 times smaller than annual cycle, deeming it negligible.

5. Lines 100-111: 'Common model error'? Is that correct? It should be 'common mode error' / 'common mode noise'.

Thank you. We substituted 'common mode error' for 'common model error' everywhere in the manuscript.

6. Lines 112-116: This paragraph probably should have been earlier, as it now interweaves the CMC description, making it inconsistent.

Please check again the manuscript. The lines you refer to do not match up. If you refer to lines 105-109 the flow is: 1) Definition of common mode error; 2) what Kreemer and Blewitt did about common mode error; and 3) what Tian and Shen did about common mode error.

7. 'Timeseries' vs 'time-series' vs 'time series'. Please be consistent throughout the manuscript.

Thank you. We use the term 'timeseries' throughout the manuscript.

8. 'GPS up displacements' vs' GPS vertical displacements'. Please be consistent throughout the manuscript.

Thank you. Fixed. The term vertical displacement is used throughout the manuscript.

9. 'data set' vs 'dataset'. Please be consistent throughout the manuscript.

Thank you. Fixed.

10. Lines 200-202: Is there any difference between terms 'position time series' and 'displacement time series' you use? If yes, please, explain. If not, please stay consistent.

Thank you for your comment. We revised our description earlier in the text to clarify.

"We input the NGL position timeseries, derive the displacement relative to a reference epoch and then follow Argus et al. (2010, 2017, 2021) to isolate the part of GPS displacements reflecting solid Earth's elastic response"

11. All station names are usually written in capital letters. Please correct in the text and drawings.

Done.

12. Line 302: I would suggest using GPS observed vertical displacement or GPS vertical displacement for short.

Thank you! We revised accordingly.

13. Line 340: no k and h in the explanation.

You probably mean line 240. We revised and explain the terms. Thanks!

14. To what degree and order did the authors determine the spherical harmonics from the mascons?

We use 3-degree mascons, which spherical harmonic expansion is up to degree and order 360.

15. Figure 2: Are correlation values plotted for the residuals of equation (2)? If so, why is a dependency between annual amplitude and correlation observed? Is there any residual annual signal in the displacement residuals?

No, for the correlation the original timeseries was used. Please see lines 278-279 "First, we specify the level of agreement between the datasets by estimating the Pearson correlation coefficient between GPS and GRACE(-FO) timeseries." Therefore, the dependency is anticipated.

16. Figure 3: is the variance estimated for the annual signal?

The variance is estimated from the fit of the GPS and GRACE-(FO) timeseries of an offset, a rate, and a sinusoid with a period of 1 year.

17. Description of Fig. 5, something is missing in the second sentence.

Thank you! $r_{MAD}$ was missing and was added.

18. Fig. 6: Are these the amplitudes of noises, or…?

Yes. This is the amplitude of the noise/uncertainty. We revised accordingly.

19. Line 696: GPS-derived or GPS-observed displacements? Please, stay consistent.

Thank you. We use the term "GPS-derived" throughout the manuscript.

20. Line 754: do you mean 'non-tidal' atmospheric and oceanic loading models?

Thanks for picking it up. It is indeed non-tidal.